# *Drosophila MESR4* Gene Ensures Germline Stem Cell Differentiation by Promoting the Transcription of *bag of marbles*

**DOI:** 10.3390/cells11132056

**Published:** 2022-06-28

**Authors:** Alexandra Brigitta Szarka-Kovács, Zsanett Takács, Melinda Bence, Miklós Erdélyi, Ferenc Jankovics

**Affiliations:** 1Institute of Genetics, Biological Research Centre, Eötvös Loránd Research Network, H-6726 Szeged, Hungary; szarka.brigitta@brc.hu (A.B.S.-K.); lakatos.zsanett@brc.hu (Z.T.); bence.melinda@brc.hu (M.B.); 2Doctoral School in Biology, University of Szeged, H-6720 Szeged, Hungary; 3Department of Medical Biology, Albert Szent-Györgyi Medical School, University of Szeged, H-6720 Szeged, Hungary

**Keywords:** *Drosophila*, ovary, germline stem cell, stem cell niche, germ cell, stem cell differentiation, bag of marbles, MESR4

## Abstract

Ovarian germline stem cells (GSCs) of *Drosophila* melanogaster provide a valuable in vivo model to investigate how the adult stem cell identity is maintained and the differentiation of the daughter cells is regulated. GSCs are embedded into a specialized cellular microenvironment, the so-called stem cell niche. Besides the complex signaling interactions between the germ cells and the niche cells, the germ cell intrinsic mechanisms, such as chromatin regulation and transcriptional control, are also crucial in the decision about self-renewal and differentiation. The key differentiation regulator gene is the bag of marbles (bam), which is transcriptionally repressed in the GSCs and de-repressed in the differentiating daughter cell. Here, we show that the transcription factor MESR4 functions in the germline to promote GSC daughter differentiation. We find that the loss of MESR4 results in the accumulation of GSC daughter cells which fail to transit from the pre-cystoblast (pre-CB) to the differentiated cystoblast (CB) stage. The forced expression of bam can rescue this differentiation defect. By a series of epistasis experiments and a transcriptional analysis, we demonstrate that MESR4 positively regulates the transcription of bam. Our results suggest that lack of repression alone is not sufficient, but MESR4-mediated transcriptional activation is also required for bam expression.

## 1. Introduction

Stem cells are undifferentiated cells possessing the unique ability of unlimited cell division capacity. Stem cells can undergo symmetric or asymmetric mitotic divisions to accomplish the dual task of self-renewal and the generation of differentiated cells [1]. The symmetric division of a stem cell can produce only stem cell daughters in some divisions and only differentiated daughters in others. This way, the number of stem cells can expand during development or regeneration while maintaining the balance between stem cells and differentiated cells. Upon asymmetric mitosis of a stem cell, however, one of its daughters remains in the stem cell state to maintain the undifferentiated stem cell pool, while the other daughter cell starts to differentiate to support developmental processes or to ensure tissue homeostasis [2]. The better understanding of the mechanisms behind the decision between self-renewal and differentiation is a key question in stem cell biology. The regulation of the *Drosophila* female germline stem cells (GSCs) provides an excellent model for studying stem cell maintenance and differentiation processes [3,4,5]. In the *Drosophila* females, a population of GSCs is located in the niches at the anterior tip of the ovary [6]. Each GSC divides asymmetrically and generates a daughter cell staying anchored at the tip of the niche and remaining in the GSC state, while the other daughter cell becomes a pre-cystoblast (pre-CB) committed to differentiate [7,8]. After mitosis, the GSC and the newly formed pre-CB remain physically connected due to an incomplete cytokinesis. The thin cytoplasmic connection is abscised in the G2 phase of the subsequent cell cycle [9,10,11]. The pre-CB then starts to differentiate and becomes a cystoblast (CB) which undergoes four synchronous, incomplete divisions, and gives rise to a sixteen-cell cyst. One cell of the sixteen-cell cyst matures to an oocyte, and the others differentiate into nurse cells. The differentiation state of the germ cells can be easily followed by visualizing specific cytoplasmic organelles. GSCs, pre-CBs and CBs contain a dot like cytoplasmic organelle, the spectrosome, while the cyst is marked by a branched structure, called the fusome (Figure 1A).

GSC self-renewal and differentiation are tightly regulated by the germ cell’s intrinsic and extrinsic factors [12]. The extrinsic factors are generated by cap cells and escort cells which surround the GSCs. These somatic cells create the stem cell niche and regulate the GSC differentiation [13,14,15]. The somatic niche cells secret Decapentaplegic (dpp): the *Drosophila* homolog of the Bone morphogenetic protein [8]. Dpp forms a short-range morphogen gradient and activates the dpp signaling pathway exclusively in the GSCs [16]. The main function of the dpp pathway in the GSCs is to repress the differentiation, and thus maintain the GSCs’ undifferentiated state [17]. Following mitosis of the GSCs, the pre-CBs are displaced from the stem cell niche losing the physical contact with the somatic cap cells, i.e., the source of the inhibitory dpp signal. At this position, the pre-CBs receive less Dpp, which leads to their differentiation into CBs [18].

Besides the extrinsic factors, the germ cell intrinsic programs are crucial to control the differentiation of the GSCs [12]. The transition from self-renewal to differentiation relies on epigenetic mechanisms, mRNA regulation and on protein metabolism. In GSCs, Nanos and Pumilio suppress the translation of mRNAs that promote GSC differentiation [19,20,21]. The miRNA-based (miRNA) silencing machinery also plays an essential role in GSC self-renewal by translational repression of differentiation factors in the GSCs [22,23,24]. In the CBs, the enhanced biogenesis of ribosomes and an increased global protein synthesis have been shown to be required for differentiation [25].

In addition to the regulation of protein turnover and translational repression, the epigenetic control of transcription is also involved in switching from the stem cell fate to the differentiated state [26,27] Epigenetic mechanisms include structural changes of the chromatin or recruitment of transcription regulators, which prevent the ectopic expression of germline differentiation genes in GSCs or activate them in pre-CBs and CBs.

The key regulator gene promoting GSC differentiation is *bam*. Pre-CBs lacking *bam* expression fail to differentiate, whereas ectopic *bam* expression in the GSCs leads to precocious differentiation and GSC loss, indicating that *bam* is both necessary and sufficient for the differentiation. These observations have led to the simple model in which silenced *bam* expression in the GSCs maintains self-renewal, and *bam* de-repression in the pre-CBs promotes differentiation [7,8,17].

In the GSCs, the expression of *bam* is suppressed by the transcription factor Mad, which is activated by phosphorylation in response to the dpp signal [17]. Phosphorylated Mad (pMad) translocates into the nucleus to directly suppress *bam* transcription. In the pre-CBs, however, turnover of pMad is shifted towards its degradation, which leads to the de-repression of *bam* transcription and to differentiation [28]. Recently, the transcription factor Krüppel (Kr) has been demonstrated to be involved in the temporal regulation of *bam* transcription independently of the dpp signaling. [29]. Krüppel suppresses *bam* transcription in the germ cells at the larval stages preventing their precocious differentiation into CBs.

Besides *trans* acting factors, *cis* regulatory elements are essential for the proper temporal control of *bam* transcription. A suppressor element, located upstream of the transcription start site, associates with Kr to maintain the repressed state of *bam* in the larval germ cells [29]. Another 18 bp long silencer element (SE) in the 5′UTR of the *bam* gene mediates the dpp-dependent suppression of *bam* transcription in the GSCs. The *bam* transcriptional regulator contains an enhancer element responsible for germ cell specific *bam* expression. In the GSCs, the Dpp signal maintains the silencer in an active state and *bam* transcription is suppressed. At the pre-CB stage, the SE becomes inactive and the *bam* transcription commences under the control of the active enhancer elements, which switches the state of the pre-CB to the CB fate [17,30].

While the negative regulation of *bam* expression is well known, the positive regulation is poorly studied. Here, we demonstrate that *MESR4* is a *trans* acting factor positively regulating *bam* transcription to promote differentiation of the GSCs. GSCs lacking *MESR4* initiate their differentiation into pre-CBs, but fail to adopt the differentiated CB fate, which leads to the accumulation of pre-CBs in the niche. We propose a model in which *MESR4* promotes *bam* expression in the pre-CBs to support the transition of the committed pre-CB to the differentiated CB state.

## 2. Materials and Methods

### 2.1. Drosophila Stocks and Genetics

Flies were raised at 25 °C unless specified otherwise. The strains used in this study include: bamP-Bam:GFP and bamPΔSE-Bam:GFP [30], *MESR4^79^* [31], bam-Gal4 and bamP-GFP [30], hs-*bam* [7], Pgc:GFP [32] nos-Gal4Vp16 [33], c587-Gal4 (Bl#67747), osk-Gal4VP16 (BL#44242), Pmatalpha4-Gal4VP16 (Bl#706), P{TRiP.HMC04881}attP40 (*MESR4*-RNAi-1 Bl#57564)), P{TRiP.GL00462}attP2 (*MESR4*-RNAi-2 Bl#35618), P{TRiP.HMS00029}attP2 (Bam-RNAi Bl#33631), bgcn^EY00974^ (Bl#20106), vasa-Cas9 (Bl#51323), *MESR4*-GFP.FPTB (Bl#67731), *bgcn^1^* (Bl#6054), Df(2R)BSC136 (Bl#9424).

The *MESR4*^Δ*PHD*^ allele was generated by CRISPR/Cas9 mediated genome engineering [34]. To generate the *MESR4*^Δ*PHD*^ allele, target sites of two sgRNAs were inserted into the pCFD4 vector (Addgene, 49411) and introduced into the attP2 docking site on the *Drosophila* genome by standard transformation methods [34] (Table 1). *MESR4*^+^; Vasa-cas9/attP2-pCFD4-*MESR4*-gRNA males were generated and the mutagenized second chromosomes were isolated. The *MESR4*^Δ*PHD*^ deletion allele was confirmed by sequencing.

### 2.2. Heat Shock

Two-day-old nosGal4 > *MESR4*-RNAi; hs-*bam* females were heat-shocked at 37 °C twice for 1 h, separated by a 2 h recovery period at 25 °C. For the control, nosGa4 > *MESR4*-RNAi; hs-*bam* flies without heat shock were used. The phenotypes were examined one day after heat shocks.

### 2.3. Immunohistochemistry

Immunostainings were performed as described earlier [35,36]. The following primary antibodies were used: anti-vasa (1:300, DSHB), anti-HTS (1:20, DSHB, 1B1), anti-GFP (1:500, Invitrogen, Waltham, MA, USA, A-11122), anti-Smad3 (1:100, abcam, ab52903), anti-cycB (1:30 DSHB, F2F4). The tissues were observed with a Leica SP5 or with Zeiss LSM800 confocal microscope.

### 2.4. Quantitative PCR

For the quantitative PCR, total RNA was prepared using the Reliaprep RNA tissue Miniprep System (Promega, Medison, WI, USA, Z6111). To ensure comparability of the ovaries, *bgcn^1^*/Df(2R)BSC136 mutant ovaries were used as a control and *MESR4* and *bam* silencing was performed on *bgcn^1^*/Df(2R)BSC136 mutant background. Ovaries were collected from 2-days-old *bgcn^1^*/Df(2R)BSC136 and nosGal4>*MESR4*-RNAi and nosGal4 > *bam*-RNAi ovaries [37]. To achieve a comparable reduction of *bam* expression in *MESR4*-RNAi and *bam*-RNAi females, nosGal4 > *bam*-RNAi females were raised at 18 °C.

OligodT primers were used to synthetize cDNAs (First Strand cDNA Synthesis kit, ThermoScientific, Waltham, MA, USA, K1612). qPCR reactions were performed using SensiFAST SYBR Hi-ROX Kit (Bioline, London, UK, BIO-92005). The thermal cycling condition consisted of 95 °C for 30 min, 95 °C for 30 s, 60 °C for 30 s, 72 °C for 30 s, 35 cycles. Melt: 60–95 °C, 1 °C each step. For each reaction, there were three technical replicates and three biological replicates. An Rp49 transcript was used for internal control (Table 2). Rotor-Gene Q Series software was used for data analysis.

## 3. Results

### 3.1. MESR4 Promotes GSC Differentiation in a Cell Autonomous Manner

*MESR4* has been shown to be required for the proper germ cell development [31]. To determine the spatial and temporal requirement of *MESR4*, we silenced it in the ovarian somatic and germ cells at distinct stages of the germ cell differentiation by expressing *MESR4*-shRNAs with tissue specific Gal4 drivers with various expression patterns in the ovary.

First, we investigated how a somatic *MESR4* function affects germline behavior. Therefore, the c587Gal4 line was used to induce *MESR4* silencing in all somatic cell types of the ovary and the differentiation state of the germ cells was analyzed. By immunolabelling the spectrosomes and the fusomes, no GSC differentiation defects were detectable in c587Gal4 > *MESR4*-shRNA niches (Appendix A). The adult c587Gal4 > *MESR4*-shRNA females were fertile, indicating that the *MESR4* function in the ovarian soma is dispensable for germ cell development.

To examine the *MESR4* function in the germline, we expressed two independent *MESR4*-shRNAs (GL00462 and HMS00029) by Gal4 drivers specifically active in the germ cells at various stages of the oogenesis. The oskGal4 and matTubGal4 drivers were used to silence *MESR4* outside of the GSC niche, at the later stages of the germ cell development. No defects were detected in oskGal4 > *MESR4*-shRNA and matTubGal4 > *MESR4*-shRNA ovaries, indicating that *MESR4* is required for the early germ cell development exclusively in the niche (Appendix A).

Next, we silenced *MESR4* by the nosGal4 driver which started to express in the GSCs, and its expression persisted throughout the oogenesis. While wild type control niches carried less than four spectrosome-containing cells (3.9 ± 0.8, n = 24) which could be GSCs, pre-CBs or CBs, the niches of the nosGal4 > *MESR4*-shRNA females contained many spectrosome-containing single germ cells (35.0 ± 10.2 in *MESR4*-RNAi line 1, n = 24 and 30.5 ± 6.1 in *MESR4*-RNAi line 2, n = 24) which formed germ cell tumors (Figure 1C,D). The accumulation of spectrosome-containing cells in the niches indicates that germ cells failed to differentiate. The silencing of *MESR4* with both *MESR4*-shRNA lines resulted in identical differentiation defects confirming the specificity of the tumorous *MESR4*-RNAi phenotype (Figure 1B). In summary, these results have revealed that *MESR4* is required for germ cell development and functions exclusively in the germline where it promotes the differentiation of the germ cells in a cell autonomous manner.

### 3.2. MESR4 Regulates Germ Cell Differentiation at the Pre-CB Stage

The accumulated undifferentiated germ cells observed in the nosGal4 > *MESR4*-shRNA niches could be GSCs which maintain their self-renewal capacity outside of the niche. Alternatively, these cells are daughter cells of GSCs, pre-CBs or CBs, which fail to complete their differentiation program. To distinguish between these possibilities, we analyzed the presence of GSC and CB specific molecular markers in the *MESR4*-RNAi niches. In the wild type, GSCs can be identified by their physical contact with the cap cells and by the presence of phosphorylated Mad (pMad), a reporter of an active dpp pathway. Similar to wild type, in the nosGal4 > *MESR4*-shRNA niches, two to three GSCs accumulated pMad in their nuclei (Figure 2A,B). NosGal4 > *MESR4*-RNAi cells located outside the GSC niche lacked nuclear pMad, indicating that these cells are not GSCs.

The hallmark of differentiated CBs is the expression of *bam* which is initiated in the GSC daughter cells that lose physical contact with the cap cells. To monitor *bam* expression, the bamP-GFP transcriptional reporter line was used which contains the *bam* promoter region fused with a GFP coding sequence [30]. In wild type niches, strong GFP expression was detected (Figure 2C). In the undifferentiated nosGal4 > *MESR4*-shRNA germ cells, however, a week *bam* expression was detected when monitored with the bamP-GFP reporter line (Figure 2D). Consistent with this observation, a significant reduction of *bam* mRNA levels was found by qRT-PCR in the nosGal4 > *MESR4*-shRNA ovaries (Figure 2E). The *bam* mRNA levels of the *MESR4*-silenced germ cells were similar to these of the tumorous ovaries of nosGal4 > *bam*-shRNA females, indicating that this degree of reduction in *bam* mRNA levels is sufficient to prevent differentiation.

Next, we silenced *MESR4* with the bamGal4 driver, which is specifically active in the differentiated CBs reflecting the expression pattern of the endogenous *bam* gene. In the bamGal4 > *MESR4*-shRNA ovaries, no tumorous niches were found, indicating that *MESR4* is not required in *bam* expressing CBs (Appendix A). Based on these data, we hypothesized that *MESR4* functions upstream of *bam* to promote GSC differentiation. To validate our argument, we performed a genetic rescue experiment using the hs-*bam* transgenic line. Forced expression of *bam* from the heat shock inducible hs-*bam* transgene induced differentiation of the *MESR4*-RNAi germ cells. Following heat shock, fusome-containing cysts were formed in the nosGal4 > *MESR4*-shRNA; hs-*bam* niches (Figure 3B,C). Forced expression of *bam* resulted in the reduction of the number of GSC-like tumors (37.0% in nosGal4 > *MESR4*-shRNA; hs-*bam*, n = 27 vs. 96.3% in nosGal4 > *MESR4*-shRNA, n = 25) (Figure 3D). In summary, *MESR4* regulates the differentiation in the committed GSC daughter cells by promoting *bam* expression.

### 3.3. MESR4 Is Required for the Transition from the Pre-CB to the CB Stage

An analysis of GSC and CB specific molecular markers of *MESR4*-silenced germ cells revealed that these cells exit the cells’ GSC state, but do not adopt the differentiated CB fate. Thus, we concluded that the silencing of *MESR4* blocks the differentiation process at the pre-CB stage. To test this hypothesis, we investigated the expression of *polar granule component* (*pgc*), specifically activated in the pre-CBs prior to the expression of the differentiation factor bam [32]. We used the pgcGFP reporter line expressing EGFP under the control of the endogenous regulatory elements of *pgc* (Figure 4A). The silencing of *bam* resulted in the arrest of the differentiation process at the pre-CB stage, as indicated by the accumulation of the pgcGFP-positive pre-CBs in the nosGal4 > *bam*-shRNA niches (Figure 4B). Similar to *bam* silencing, the silencing of *MESR4* resulted in the accumulation of the pgcGFP-positive pre-CBs in the nosGal4 > *MESR4*-shRNA niches, indicating that *MESR4* promotes the transition from pre-CBs into the CB stage (Figure 4C).

Cell cycle control is tightly associated with the regulation of the pre-CB-to-CB transition. In the pre-CBs, Pgc indirectly promotes the accumulation of Cyclin B (CycB) at the G2 phase. Then, CycB activates the mechanisms leading to the expression of *bam* and the transition to the CB stage [32]. We correlated the phase of the cell cycle and the transition state from pre-CBs into the CB stage with the requirement of the *MESR4* function. We investigated the expression of the cell cycle regulator CycB, a marker of late G2, in *MESR4*-silenced germ cells by immunostaining (Figure 4D) [32]. We observed a robust accumulation of CycB-positive germ cells in the nosGal4 > *bam*-shRNA niches (Figure 4E). Similar to *bam* silencing, undifferentiated nosGal4 > *MESR4*-shRNA germ cells were found to express CycB (Figure 4F). The accumulation of late G2 phase pre-CBs in the niches indicates that *MESR4* functions downstream of *pgc* and CycB to promote *bam* expression. In summary, we conclude that *MESR4* is required in the pre-CBs immediately before *bam* expression to drive the transition of the undifferentiated G2 phase pre-CBs into the differentiated CB state.

### 3.4. MESR4 Promotes the Transcription of Bam

To identify the molecular mechanisms by which *MESR4* regulates GSC differentiation, we first determined the expression pattern of GFP-tagged *MESR4* expressed from a genomic bacmid construct (*MESR4*:GFP) [38]. The tagged *MESR4* variant completely rescued the lethal phenotype associated with the *MESR4^79^* mutation, indicating that the MESR4:GFP fusion protein is fully functional. We detected ubiquitous *MESR4* expression in the somatic and germ cells of the ovaries of *MESR4*:GFP females. In the cells, *MESR4* localized to the nuclei, suggesting a role for *MESR4* in the transcriptional regulation of *bam* (Figure 5A).

The *bam* transcription has been shown to be negatively regulated by the dpp signaling pathway through the SE element of the *bam* regulator region. In the bamPΔSE-Bam:GFP transgenic females, the SE element is removed and the GSCs do not respond to the inhibitory pMad-mediated Dpp signal and initiate *bam* expression from the transgene. Ectopically expressed bam then leads to the differentiation of the GSCs into CBs and cysts, which in turn results in “empty” niches containing no GSCs (Figure 5B). To investigate whether the forced GSC differentiation caused by the de-repressed expression of *bam* can be rescued by the loss of *MESR4*, we used bamPΔSE-Bam:GFP transgenic females ectopically expressing *bam* in the GSCs and simultaneously silenced *MESR4* in the germline. We found that in the bamPΔSE-Bam:GFP; nosGal4 > *MESR4*-shRNA ovaries the number of empty niches was reduced compared with bamPΔSE-Bam:GFP control flies (19.5%, n = 87 vs. 56.6%, n = 99) (Figure 5B–E). Furthermore, bamPΔSE-Bam:GFP; nosGal4 > *MESR4*-shRNA niches accumulated spectrosome containing single cells (37.9%, n = 87 vs. 0.0%, n = 99) (Figure 5B–E). Taken together, the suppression of GSC-loss phenotype by the silencing of *MESR4* indicates that the *MESR4* acts as antagonistic to the dpp-mediated *bam* suppression. In the pre-CBs, the lack of the dpp-mediated suppression is not sufficient, but also a *MESR4*-mediated positive regulation is required for the initiation of *bam* expression.

### 3.5. The PHD Domain Is Dispensable for MESR4 Function

An in silico analysis of *MESR4* revealed that the *MESR4* protein possesses a plant homeodomain (PHD) finger generally involved in chromatin remodeling nine C2H2 type zinc finger (ZF) domains mediating sequence specific DNA–protein interactions [39]. Due to its domain composition, *MESR4* can control the *bam* expression by several, not mutually exclusive, mechanisms. By the PHD finger, it may act as a chromatin remodeling factor and promote *bam* expression by the formation of a permissive chromatin environment. Alternatively, *MESR4* can function as a DNA-binding transcription factor associating directly with cis-regulatory elements of differentiation promoting genes by the ZF domains. To investigate how *MESR4* controls the *bam* expression, we analyzed the activity of the *MESR4* PHD finger domain using a CRISPR-based loss-of-function genetic approach.

PHD fingers have been reported to be involved in epigenetic gene regulation by mainly recognizing various unmodified or methylated lysines of H3 histone molecules [40]. To test the biological relevance of the PHD finger of *MESR4*, we applied CRISPR/Cas9-mediated gene editing to generate *MESR4* alleles lacking this domain. This way, we isolated a *MESR4^Δ^*^PHD^ allele lacking the C-terminal 106 amino acids of *MESR4* covering the entire PHD finger. The homozygous mutant *MESR^Δ^*^PHD^ animals were viable, indicating that the PHD finger of *MESR4* is not required for viability. The homozygous females were fertile, and wild type fusome containing cysts were detected in their niches. A lack of germ cell differentiation defects indicates that the PHD finger is dispensable for the *MESR4* function in *bam*-mediated pre-CB to CB transition.

## 4. Discussion

The mechanisms behind the decision between self-renewal and differentiation act at multiple levels of gene regulation. Transcriptional control, regulation of protein or RNA turnover and translational regulation of differentiation genes act in concert to ensure the proper temporal and spatial regulation of GSC maintenance and differentiation. Here, we show that *MESR4* controls the fate of germ cells in a very narrow developmental window by promoting the pre-CB to CB transition, and functions as a positive transcriptional regulator of the differentiation gene *bam*.

At the transcriptional level, GSC differentiation can be theoretically triggered by the repression of self-renewal factors, by preventing the expression of differentiation inhibitory factors or by stimulating the expression of differentiation factors. In these processes, epigenetic mechanisms, such as the incorporation of histone variants, posttranslational histone modifications and repositioning of nucleosomes by chromatin remodelers play a key role by establishing a global chromatin landscape to control transcription in GSCs and in their differentiating daughter cells [26]. In the GSCs, epigenetic mechanisms intrinsically suppress the transcription of *bam* to prevent the differentiation [41,42,43,44,45,46]. In the pre-CBs, however, the chromatin-dependent transcriptional suppression of *bam* is released to trigger the transition of the pre-CBs to the differentiated CBs. While the mechanisms mediating the transcriptional suppression of *bam* in the GSCs are well known, the identity of positive transcriptional regulators that drive *bam* expression to facilitate the differentiation of the GSC daughter pre-CBs remains elusive. We propose that *MESR4* is such a factor which functions in the pre-CB to promote *bam* transcription.

Although *MESR4* is expressed in the germ cells and in all somatic cell types of the GSC niche, its function is exclusively required cell-autonomously in the germ cells at the pre-CB stage, as demonstrated by the accumulation of undifferentiated pre-CBs upon germline-specific *MESR4* silencing. How the activity of *MESR4* is spatially and temporally regulated remains elusive. *MESR4* may function as a component of a protein complex specifically assembled in the pre-CBs and may bind a yet undetermined factor, the expression of which is restricted to the pre-CBs. Alternatively, *MESR4* may be activated by upstream processes specifically taking place in the pre-CBs. In the pre-CBs, the transcriptional repressor Pgc is transiently expressed mediating a pulse of global transcriptional silencing which prevents the transcription of stem cell fate regulators [32]. This is accompanied by the resetting of chromatin modifications which generates an epigenetic landscape competent to express determinants of a different cell fate. This chromatin environment may permit *MESR4* to drive *bam* expression directly or indirectly.

The *MESR4* function has been implicated in various developmental processes as a downstream effector of signaling pathways. It is able to suppress the constitutively activated Ras-MAPK pathway [47] or the FGF-signaling [48]. In addition, *MESR4* is required for the proper control of EGFR/ERK signaling during embryonic development and wing formation [49]. In the adult female GSC niche, the EGFR/ERK pathway is active in the escort cells, and limits the Dpp gradient to the anterior tip of the niche where the GSCs reside [50]. Several lines of evidence indicate that in this cellular context *MESR4* does not function as a component of the EGFR/ERK pathway. Impaired EGFR signaling in the niche results in the accumulation of GSC-like cells induced by the expanded dpp activity, whereas silencing of *MESR4* in the germ cells induces the accumulation of pre-CBs. The *MESR4* function is dispensable for proper germ cell development in the escort cells where the EGFR pathway is active.

Our results suggest that the molecular mechanism by which *MESR4* regulates germ cell development is the transcriptional control of *bam*. Based on its protein domain composition, *MESR4* may promote *bam* transcription as a chromatin remodeling factor via the PHD finger domain [51,52]. PHD fingers are found in a number of chromatin remodeling factors involved in nucleosome binding where PHD domains interact with modified histones [53,54,55,56]. However, we found that the PHD finger is dispensable for the *MESR4* function in germ cell differentiation. Consistent with our observation, previous studies failed to detect the direct association of *MESR4* with the modified histones [49]. While we cannot completely exclude that *MESR4* is involved in chromatin remodeling, we prefer an alternative hypothesis for the *MESR4* function. According to this hypothesis, *MESR4* directly interacts with the DNA via its Zn-fingers to specifically control the transcription of key differentiation regulators. An obvious candidate for a target of *MESR4* regulation could be *bam*. This hypothesis is supported by two lines of evidence. We detected decreased *bam* mRNA levels in *MESR4*-silenced ovaries. Furthermore, the phenotype caused by the unsilenced *bam* expression from the bamΔSE transgene can be suppressed by silencing *MESR4*, thus confirming that *MESR4* promotes *bam* expression. However, the precise mechanisms of *bam* regulation by *MESR4* remain elusive. MESR4 may indirectly affect *bam* expression through the transcriptional regulation of a yet unknown factor. Alternatively, the transcriptional control of *bam* by MESR4 may be mediated directly by the association of MESR4 with the *bam* regulatory sequences. However, further studies are needed to distinguish between these possibilities.

## Figures and Tables

**Figure 1 cells-11-02056-f001:**
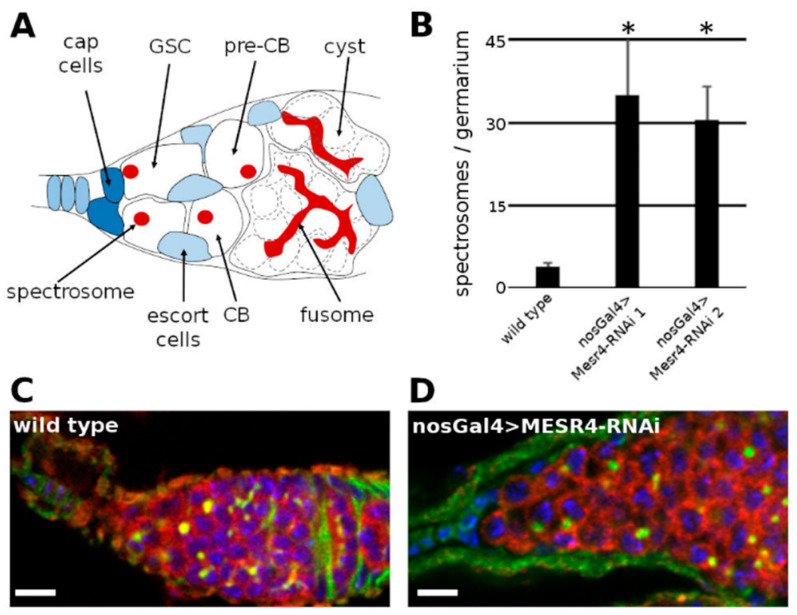
*MESR4* cell-autonomously promotes germ cell differentiation. (**A**) Schematic representation of a *Drosophila* niche where the germ cells (white) are surrounded by somatic cells (blue). The germline stem cells (GSCs) reside in the anterior tip of the niche in a specific somatic microenvironment, the stem cell niche, maintained by somatic cells. The GSCs divide continuously and give rise to a self-renewing daughter stem cell and a differentiating daughter cell, called pre-cystoblast (pre-CB). The pre-CB starts to express differentiation factors and becomes cystoblast (CB). The CB undergoes incomplete cell divisions to give rise to 2, 4, 8 and 16-cell cysts. The GSCs, pre-CBs and CBs contain a dot like structure called spectrosome (red) and the more differentiated cysts contain a branched fusome (red). (**B**) Quantification of single, spectrosome containing cells in wild type and *MESR4* silenced niches using two different RNAi lines (35.0 ± 10.2 in *MESR4*-RNAi line 1, n = 24 and 30.5 ± 6.1 in *MESR4*-RNAi line 2 n = 24 compared to 3.9 ± 0.8 in wild type control, n = 24). Data are mean ± s.d.; *t*-test, * *p* < 0.05. (**C**,**D**) Immunostaining of wild type (**C**) and nosGal4 > *MESR4*-RNAi (**D**) niche with germ cell tumor. Spectrosomes and fusomes are labelled with HTS (green), germ cells are labelled for Vasa (red); DAPI is blue. Scale bars are 10 µm.

**Figure 2 cells-11-02056-f002:**
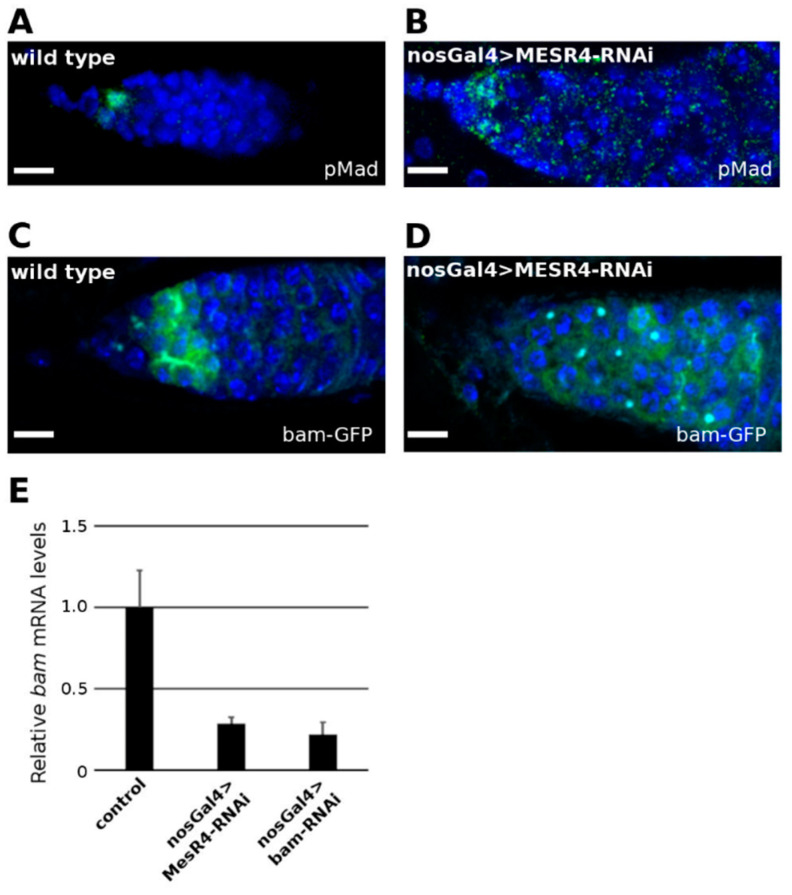
*MESR4* promotes *bam* expression in the GSC daughter cells. (**A**,**B**) Wild type control (**A**) and nosGal4 > *MESR4*-RNAi niches (**B**) stained with pMad (green) and DAPI (blue). Germline-depleted *MESR4* niches do not accumulate pMad positive cells. Scale bars: 10 µm. (**C**,**D**) Wild type control (**C**) and nosGal4 > *MESR4*-RNAi niches (**D**) expressing GFP under the control of the *bam* promoter. Niches were stained with GFP (green), and DAPI (blue). In the germline depleted *MESR4* niches there are less GFP signals. Scale bars: 10 µm. (**E**) Quantitation of steady state *bam* mRNA levels in *bgcn^1^*/Df(2R)BSC136 control, nosGal4 > *MESR4*-RNAi and nosGal4 > *bam*-RNAi ovaries. Compared to the control (1.0 ± 0.2), *bam* mRNA levels are decreased in the *MESR4* silenced (0.28 ± 0.03) and *bam* silenced (0.22 ± 0.07) ovaries.

**Figure 3 cells-11-02056-f003:**
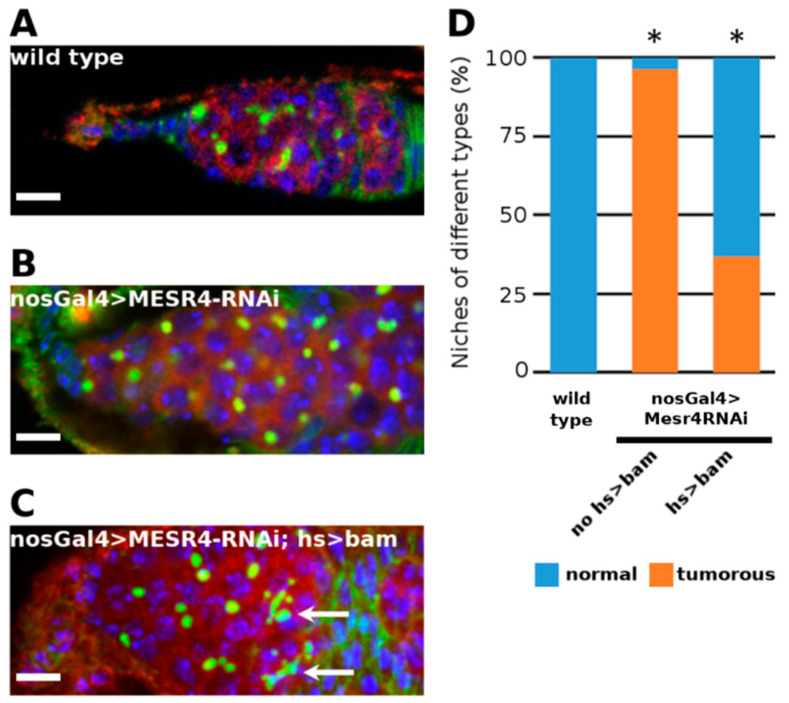
Rescue of *MESR4* silencing by forced *bam* expression. (**A**–**D**) Immunostaining a wild type (**A**) and a tumorous nosGal4 > *MESR4*-RNAi niche (**B**). Rescue of the niche defects in a nosGal4 > *MESR4*-RNAi; hs > *bam* niche after heat shock (**C**). Fusomes of differentiated germ cells are indicated by white arrows. Spectrosomes and fusomes are labelled with HTS (green); germ cells are labelled for Vasa (red); DAPI is blue. Scale bars: 10 µm. (**D**) Quantification of niche phenotypes of wild type (n = 27), nosGal4 > *MESR4*-RNAi (n = 25) and nosGal4 > *MESR4*-RNAi; hs > *bam* (n = 27) niches. “Normal” niches contain 1 to 6, “tumorous” niches contain more than 6 spectrosome containing germ cells. For statistical analysis, the Chi-square test was used, * *p* < 0.05.

**Figure 4 cells-11-02056-f004:**
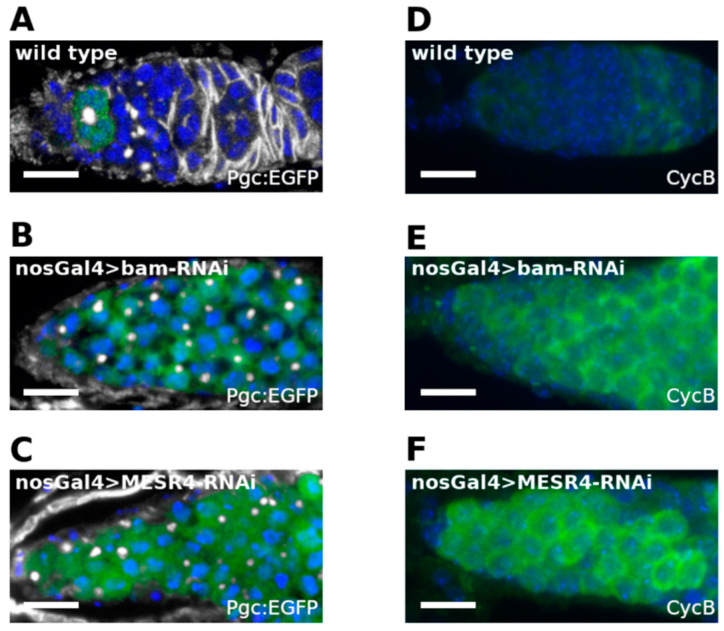
Analysis of pre-CB specific molecular markers of MESR4-silenced germ cells. (**A**–**C**) Immunostaining of niches expressing Pgc:EGFP. Wild type (**A**), nosGal4 > *MESR4*-RNAi (**B**) and nosGal4 > *bam*-RNAi niches are stained with GFP (green), HTS (white) and DAPI (blue). Niches silenced for *MESR4* or *bam* accumulate a high number of Pgc-positive pre-CBs. (**D**–**F**) Immunostaining of niches with CycB (green) and DAPI (blue). Wild type niche (**D**). NosGal4 > *bam*-RNAi (**E**) and nosGal4 > *MESR4*-RNAi (**F**) niches accumulate a high number of CycB positive cells.

**Figure 5 cells-11-02056-f005:**
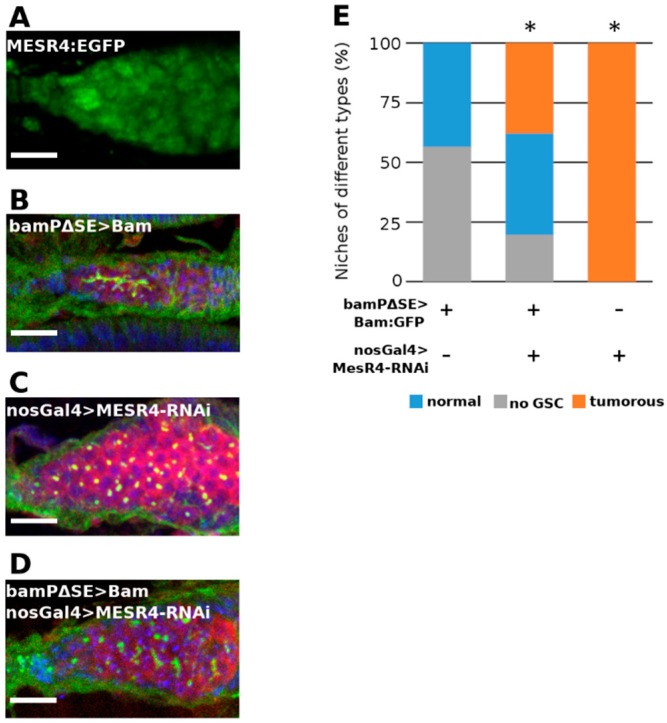
*MESR4* silencing suppresses the GSC-loss phenotype caused by de-repressed *bam* expression. (**A**) Immunostaining showing localization of *MESR4*:EGFP in the niche. *MESR4* localizes in the nuclei of the somatic cells and in the nuclei of the germ cells. Scale bar: 10 µm. (**B**–**D**) Immunostaining of niches expressing transgenic bam under the control of the *bam* promoter lacking the pMad responsive SE silencing element. Niches were stained with DAPI (blue), vasa (red) and HTS (green). Scale bars: 10 µm. In bamPΔSE-Bam:GFP niches (**B**) GSCs were lost; in nosGal4 > *MESR4*-RNAi (**C**) niches GSC-like tumors were formed. In bamPΔSE-Bam:GFP; nosGal4 > *MESR4*-RNAi niches (**D**) the number of niches without GSCs were reduced. (**E**) Quantification of germ cell differentiation phenotypes of bamPΔSE-Bam:GFP (n = 99), bamPΔSE-Bam:GFP; nosGal4 > *MESR4*-RNAi (n = 87), and nosGal4 > *MESR4*-RNAi (n = 175) niches. Niches of the “no GSC” category lacked spectrosome containing germ cells, “normal” niches contain 1 to 6, “tumorous” niches contain more than 6 spectrosome containing germ cells. For statistical analysis, the Chi-square test was used, * *p* < 0.05.

**Table 1 cells-11-02056-t001:** List of sgRNAs used for MESR4 mutant generation.

Primer	Title 2
MESR4_top	GGTAGCAGCGCTGGTGGTAT
MESR4_bottom	TGGCGATCGTCTAAGGATAAC

**Table 2 cells-11-02056-t002:** List of primers used for RT-qPCR.

Primer	Sequence
Bam_fwd	CTGCACGGCGATTGCTTAGA
Bam_rev	GTGATCATGCAGGGATCTGAAC
Rp49_fwd	CCGCTTCAAGGGACAGTATCTG
Rp49_rev	ATCTCGCCGCAGTAAACGC

## Data Availability

The datasets generated and/or analyzed during the current study are available from the corresponding author upon request.

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
