# Peer review of "Drosophila MESR4 Gene Ensures Germline Stem Cell Differentiation by Promoting the Transcription of bag of marbles"

_cells, 2022, doi:10.3390/cells11132056_

Round 1

Reviewer 1 Report

This manuscript revealed that MESR4 is required in pre-CB for pre-CB to CB transition and is required for Bam expression. The data are solid and the manuscript is well organized.

Minor concerns

1. Line 278 mentions “ mesr4 functions downstream of pgc and cycB with fig4. To me, fig4 suggests that mesr4 is upstream negative regulator of pgc and cycB.

2. The data shows that mesr4 is required for transcription of bam. I was wondering whether Bam expression is reduced with bam-Gal4 driving mesr4 shRNA. Why mesr4 knockdown with bam-Gal4 driver does not affect CB differentiation ?

3. There are a few places need correction of English senstence.

line 148, line 301,

Author Response

Minor concerns:

  1. Line 278 mentions “ mesr4 functions downstream of pgc and cycB with fig4. To me, fig4 suggests that mesr4 is upstream negative regulator of pgc and cycB.

Though the precise mechanisms regulating pgc expression and restricting it to the pre-CBs are not known, we think that MESR4 cannot be an upstream negative regulator of pgc. Loss of function of a putative negative regulator would result in pgc overexpression. (de Las Heras et al. 2009, ) demonstrated that females overexpressing pgc by the 6x[pgc] transgene are able to lay eggs indicating that upregulation of pgc expression per se does not result in tumor formation observed upon MESR4 silencing. In concert with this, (Flora et al. 2018) showed that not the elevated rather the reduced pgc function results in an increased number of undifferentiated germ cells. We have used Pgc and cycB expression as markers to determine the differentiation state of the germ cells silenced for MESR4. The only cells expressing pgc in the niche are germ cells double negative for pMAD and bam, and are therefore considered pre-CBs.

References:

Flora, Pooja, Sean Schowalter, SiuWah Wong-Deyrup, Matthew DeGennaro, Mohamad Ali Nasrallah, and Prashanth Rangan. 2018. ‘Transient Transcriptional Silencing Alters the Cell Cycle to Promote Germline Stem Cell Differentiation in Drosophila’. Developmental Biology 434 (1): 84–95. https://doi.org/10.1016/j.ydbio.2017.11.014.

Las Heras, José Manuel de, Rui Gonçalo Martinho, Ruth Lehmann, and Jordi Casanova. 2009. ‘A Functional Antagonism between the Pgc Germline Repressor and Torso in the Development of Somatic Cells’. EMBO Reports 10 (9): 1059–65. https://doi.org/10.1038/embor.2009.128.

  1. The data shows that mesr4 is required for transcription of bam. I was wondering whether Bam expression is reduced with bam-Gal4 driving mesr4 shRNA. Why mesr4 knockdown with bam-Gal4 driver does not affect CB differentiation?

Loss of function of MESR4 results in the lack of preCB-to-CB transition leading to the accumulation of cells expressing pre-CB markers pgc and cycB. Expression of wild type bam gene as well as the bamGal4 transgene are detectable only in the CBs. Thus, germ cells expressing bam and bamGal4 are differentiated cystoblasts and do not require MESR4 function any more. Accordingly, we did not find an accumulation of undifferentiated cells in the bamGal4>MESR4RNAi niches, indicating that bam expression is not reduced in these cells.

  1. There are a few places need correction of English senstence.

We have corrected the mistakes in line 148 and in line 301. We made 11 additional grammatical corrections in the manuscript.

Reviewer 2 Report

This study provides good evidence for a role of MESR4 in promoting pre-cystoblast differentiation via upregulating bam expression.

The manuscript is generally well written with some minor grammatical issues. 

I would like to see one addition to the introduction. In line 30 the authors discuss that mitosis of a stem cell results in one daughter being maintained in the stem cell poll and the other becoming committed to differentiation. I think that this should be rewritten as it only refers to stem cells that divide asymmetrically (as do the stem cells of the Drosophila germline).

The results in Figure 1 are clear. In figure 2 the authors describe the bam mRNA levels in nos>MesR4-RNAi as being similar to the tumorous ovaries of nos>bam-Rnai flies. I agree with their conclusion as depicted in panel 2E but would ask that an image of the bam-RNAi tumorous phenotype also be displayed.

The comparison between nos>MesR4-RNAi and nos>MesR4-RNAi, hs-bam clearly demonstrate that expression of bam from an exogenous promoter can rescue the MesR4 knockdown phenotype. I am not sure that the comparison to bgcn adds to the story as the comparison has been conducted using a different expression system to bam (hs vs UAS).

Could the authors please clarify the method they used for counting pre-Cb/niche in Figure 4. If they are counting green cells derived from Pgc:EGFP then the numbers of green cells shown in Panel C appear to be far in excess of the numbers reported in Panel D.

Again in Figure 5 a description of "normal" is warranted. Do the authors mean that germaria only exhibited spectrosomes near the niche and showed progressive development of fusomes?

The last paragraph of the discussion should be modified to state that the effect of MesR4 on bam transcription could be indirect and a direct effect will need to be confirmed via demonstration that MesR4 can bind the bam promoter. 

Apart from the clarifications noted above this manuscript clearly shows a new role for MesR4 in regulating cystoblast differentiation and adds to the complex nature of bam transcriptional regulation.

Author Response

The manuscript is generally well written with some minor grammatical issues.

We made 13 grammatical corrections in the manuscript.

1.

I would like to see one addition to the introduction. In line 30 the authors discuss that mitosis of a stem cell results in one daughter being maintained in the stem cell poll and the other becoming committed to differentiation. I think that this should be rewritten as it only refers to stem cells that divide asymmetrically (as do the stem cells of the Drosophila germline).

We thank the reviewer for the suggestion. We have revised the introduction and inserted three sentences explaining the difference between symmetric and asymmetric stem cell division strategies.

2.

The results in Figure 1 are clear. In figure 2 the authors describe the bam mRNA levels in nos>MesR4-RNAi as being similar to the tumorous ovaries of nos>bam-Rnai flies. I agree with their conclusion as depicted in panel 2E but would ask that an image of the bam-RNAi tumorous phenotype also be displayed.

Silencing of both MESR4 and bam results in tumorous niches. The bam loss of function phenotype is very well known and we show germ cell tumors of nosGal4>bam-RNAi females in Figure4B and E. We think, therefore that insertion of an additional image displaying nosGal4>bamRNAi niches is not necessary.

3.

The comparison between nos>MesR4-RNAi and nos>MesR4-RNAi, hs-bam clearly demonstrate that expression of bam from an exogenous promoter can rescue the MesR4 knockdown phenotype. I am not sure that the comparison to bgcn adds to the story as the comparison has been conducted using a different expression system to bam (hs vs UAS).

We agree with the reviewer. We have deleted the Figure3D and modified Figure3E to remove the experiment demonstrating that bgcn overexpression cannot rescue MESR4 silencing. We have removed the description of the experiment from the figure legend (line247) and from the main text of the manuscript. (lines239-240)

4.

Could the authors please clarify the method they used for counting pre-Cb/niche in Figure 4. If they are counting green cells derived from Pgc:EGFP then the numbers of green cells shown in Panel C appear to be far in excess of the numbers reported in Panel D.

In Figure4, we intended to demonstrate the exact differentiation stage of the tumor forming cells in MESR4 silenced ovaries. Figure4C and G clearly shows that all of the tumor forming cells express pgc and cycB indicating that the development of the germ cells stalled at the pre-CB state. For the exact comparison to the bam-silenced ovaries, we counted the pgc positive cells in the niches i.e. in the 1 and 2a region of the mutant and wild type germaria. Indeed, in contrast to the wild-type situation in the case of the tumorous germaria, delimitation of 1 and 2a regions is ambiguous, as shown by the high standard deviance in the Figure 4D. Therefore, we removed Figure4D and modified the text accordingly.

5.

Again in Figure 5 a description of "normal" is warranted. Do the authors mean that germaria only exhibited spectrosomes near the niche and showed progressive development of fusomes?

We have clarified the quantification categories shown on Figure3E and Figure5E (Figure5D in the revised version). We have inserted the description into the figure legend of Figure3 and Figure5.

6.

The last paragraph of the discussion should be modified to state that the effect of MesR4 on bam transcription could be indirect and a direct effect will need to be confirmed via demonstration that MesR4 can bind the bam promoter.

We have rephrased the last paragraph of the discussion by modifying the statement that bam could be an “obvious candidate for a direct target for MESR4 regulation”. We inserted four sentences to explain that MESR4 either indirectly or directly regulates bam transcription and these options are equally likely.